# Nanomechanical motion transduction with a scalable localized gap plasmon architecture

Brian J. Roxworthy[1] & Vladimir A. Aksyuk[1]

Plasmonic structures couple oscillating electromagnetic fields to conduction electrons in noble metals and thereby can confine optical-frequency excitations at nanometre scales. This confinement both facilitates miniaturization of nanophotonic devices and makes their response highly sensitive to mechanical motion. Mechanically coupled plasmonic devices thus hold great promise as building blocks for next-generation reconfigurable optics and metasurfaces. However, a flexible approach for accurately batch-fabricating high-performance plasmomechanical devices is currently lacking. Here we introduce an architecture integrating individual plasmonic structures with precise, nanometre features into tunable mechanical resonators. The localized gap plasmon resonators strongly couple light and mechanical motion within a three-dimensional, sub-diffraction volume, yielding large quality factors and record optomechanical coupling strength of $2\,\mathrm{THz}\cdot\mathrm{nm}^{-1}$. Utilizing these features, we demonstrate sensitive and spatially localized optical transduction of mechanical motion with a noise floor of $6\,\mathrm{fm}\cdot\mathrm{Hz}^{-1/2}$, representing a 1.5 orders of magnitude improvement over existing localized plasmomechanical systems.

[1] Center for Nanoscale Science and Technology, National Institute of Standards and Technology, 100 Bureau Drive, Gaithersburg, Maryland 20899, USA. Correspondence and requests for materials should be addressed to B.J.R. (email: brian.roxworthy@nist.gov).

Chip-scale devices coupling light with mechanical degrees of freedom are ubiquitous, finding applications from spatial light modulators in movie theaters, through telecommunications and adaptive optics for astronomy[1,2], to atomic-scale mass sensing and sensitive optical readout of micromechanical sensors such as atomic force microscopy cantilevers[3–5]. Dielectric cavity optomechanical devices with narrow resonance linewidths achieve strong coupling of optical modes with motion that is significantly influenced by the optical forces[6]. This dynamic back-action has been used to remarkable effect, such as optical cooling and observation of macroscopic quantum mechanical behaviour[7]. However, dielectric devices such as photonic crystal and whispering gallery resonators have micron-sized physical extents and minimum optical mode sizes approximately equal to the wavelength of light within the material[8]. The inability of dielectric devices to concentrate light beyond this limit hinders their usage for detecting motion from nanometre-sized regions.

Plasmonic systems offer a complimentary application domain enabled by large optical bandwidth and extreme miniaturization of optical modes. Localized plasmon resonances, supported by noble metal nanostructures, confine optical-frequency excitations into nanometre volumes by converting free-space electromagnetic energy into the electromechanical energy of charge-density oscillations[9]. The resonant frequencies of these modes depend sensitively on the nanostructures' geometric configuration such as the gap size between interacting elements[10,11]. Accordingly, strong coupling of light to nanomechanical motion can be realized by introducing a mechanical degree of freedom into the plasmonic structure[12–16]. Plasmomechanical systems based on this principle have been suggested for applications in reconfigurable metamaterials[12], compact optical switches[13] and sensors, enabling optical motion readout in nanoelectromechanical systems (NEMS) as well as dynamic light manipulation by NEMS[17]. However, to realize the benefits of this approach compared with stationary plasmonic devices and metasurfaces[18], individual plasmomechanical elements must efficiently transform minute mechanical motions into changes in far-field optical amplitude or phase. This requirement necessitates reliable production of plasmonic elements having precise, movable nanoscale gaps over a region that is large compared with the individual element size. Concurrently, the flexibility to tune the plasmonic and mechanical geometries is desirable for shaping the near-field plasmomechanical interaction. Such a scalable and flexible approach for making these devices near the physical limits of optical confinement and plasmomechanical coupling[19] is currently lacking.

In this article, we introduce a monolithic plasmonic-NEMS (pNEMS) device architecture, in which localized gap plasmon (LGP) resonators with precise, large-area nanoscale gaps are embedded into arrays of moving silicon nitride nanostructures. Our fabrication approach yields thousands of devices per chip with individually tailorable plasmonic and mechanical designs, and is compatible with optical lithography batch-fabrication and integration with electronics. The LGP resonators produce record optomechanical coupling strengths of $2\,THz \cdot nm^{-1}$ and large plasmonic quality factors. Altogether, these features enable measurement of mechanical motion from a $165 \times 350\,nm^2$ device area with a $6\,fm \cdot Hz^{-1/2}$ noise floor, representing a sensitivity that is 1.5 orders of magnitude lower than previously achieved with localized plasmonic resonators[17]. This architecture paves the way for advances in nanomechanical sensing, ultrasmall reconfigurable photonics and randomly addressable metamaterials.

## Results

**Device principle and fabrication.** In pNEMS a narrow gap between two planar horizontal gold surfaces is formed by a top rectangular prism (cuboid) and an underlying pad (Fig. 1a). The prism is embedded inside and moves with a silicon nitride ($SiN_x$) NEMS device, which in this work comprises either a $5\,\mu m$ long cantilever clamped at one end (Fig. 1b) or an $8\,\mu m$ long doubly clamped beam (Fig. 1c). We place the prism $1.5\,\mu m$ and $2.0\,\mu m$

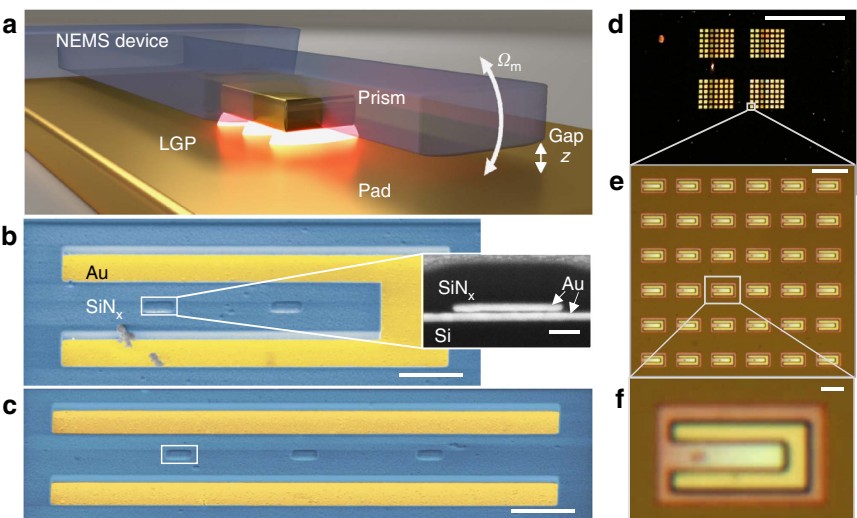

**Figure 1 | pNEMS illustration and micrographs.** (**a**) Illustration (not to scale) of a pNEMS device showing the major components of the system. The fundamental mechanical frequency is $\Omega_m$. (**b**) False-colour scanning electron microscopy (SEM) image of the cantilever. The inset is a cross-section image of the cantilever showing the embedded prism; the cross section is created using a focused ion beam. (**c**) False-colour SEM image of the beam with studied prism indicated with a white box. Scale bars, $1\,\mu m$ (inset, $100\,nm$). Prisms located further from the base in both devices are not used in this work. The prisms are confined within the mechanical resonators upon release, and, therefore, travel with the structures as they move, as illustrated by the presence in the cross section of a small gap separating the prism from the pad after the cantilever is stuck to the pad due to charging in the SEM. Micrographs are made after completing all the optomechanical measurements; debris on cantilever accumulated after all measurements were taken. (**d–f**) Optical micrographs of the pNEMS chip at successive zoom-in levels spanning three orders of magnitude in size; scale bars are (**d**) $1\,mm$, (**e**) $10\,\mu m$ and (**f**) $1\,\mu m$.

from the base of the cantilever and beam, respectively. An LGP resonance forms in the nominally 12 nm gap under the 35 nm thick prism, with a resonator footprint defined by the 350 nm length and 165 nm width. The fabrication process flow is given in Supplementary Fig. 1. Most of the LGP total energy is located within the gap, which results both in strong coupling between plasmonic and mechanical modes, and large plasmonic quality factors due to low radiative loss.

Devices are produced with three sequential steps of aligned electron beam lithography. Gold pads and prisms are shaped by metal evaporation and liftoff, while dry etching is used for the $SiN_x$ mechanical components (Methods, Supplementary Note 1). We use a uniform Cr sacrificial layer to define the LGP resonator gap. Cr is selectively removed with a wet-chemical etch to release the $SiN_x$ nanomechanical structures, freeing them to move. This method of defining the gap makes narrow and very large aspect ratio gaps possible. Low-temperature (180 °C) plasma-enhanced chemical vapour deposition is used to deposit a conformal $SiN_x$ structural NEMS layer directly on top of the plasmonic elements. This deposition embeds the prism within the $SiN_x$ while avoiding morphological changes that can result from surface melting below the bulk melting point of gold[20]. Owing to the simple lithographic approach, large arrays of pNEMS (Fig. 1d–f) with a broad range

of different device designs are fabricated simultaneously (Supplementary Fig. 2).

**Characterizing LGP modes.** The plasmonic response of the pNEMS is characterized by confocal spectroscopy on individual LGP resonators (Methods, Supplementary Note 2, Supplementary Fig. 3). Figure 2a reveals the LGP modes as pronounced dips in the reflectivity spectra. Fitting these data to Lorentzian curves, we find resonant wavelengths ($\lambda_{LGP}$) of 790 nm and 760 nm and quality factors ($Q_{LGP}$) of $23 \pm 0.6$ and $21.6 \pm 1.7$ for the cantilever and beam, respectively; uncertainties refer to s.e. derived from the fitting procedure. The measured spectra are in good agreement with the reflectivity calculated using a three-dimensional (3D), full-vector finite-element frequency-domain computation (Fig. 2b); see Methods, Supplementary Note 3. Differences in the measured linewidths and experimentally determined values of $Q_{LGP}$ for the two devices, as well as their slight deviation from calculated values, are attributed to fabrication inhomogeneities such as variation in the prism shape. Here $\lambda_{LGP}$ refers to the free-space probe wavelength at which the LGP mode appears in the measured and calculated reflectivity curves in Fig. 2. However, the physical size of the LGP mode is reduced to $\lambda_{LGP}/n_{eff}$, where $n_{eff}$ is

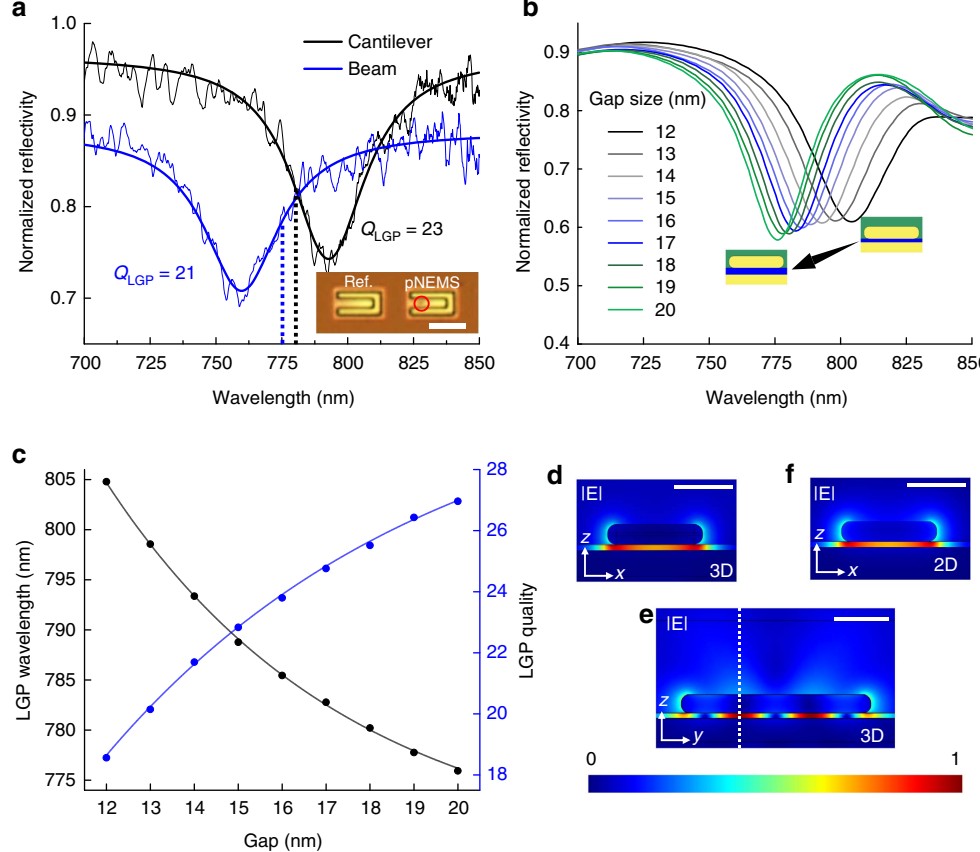

**Figure 2 | Tunable localized gap plasmon resonators.** (**a**) Measured LGP spectral responses. The shorter $\lambda_{LGP}$ in the case of the beam is expected due to a larger gap as fabricated. In both cases, the final gap is larger than the as-deposited 12 nm Cr layer; (Supplementary Notes 1 and 5). The dotted lines represent the laser wavelengths of 775 nm and 780 nm used subsequently for motion transduction of the beam and cantilever, respectively. The inset shows an optical image of a typical pNEMS device, prism location marked with a red circle, adjacent to a reference device; scale bar, 5 μm. (**b**) Calculated reflectivity for the pNEMS as a function of gap size. (**c**) Plasmonic quality factor and LGP wavelengths extracted from Lorentzian fits of theoretical reflectivity curves. Solid lines are fits to exponential functions. Reflectivity curves are normalized to those (**a**) measured and (**b**) calculated for a reference pNEMS devices without a prism. Normalized value of the LGP total electric field on a plane through the prism width calculated using (**d**) 3D full-field and (**f**) two-dimensional (2D) eigenmode methods. (**e**) 3D normalized electric field displayed on a plane through the prism length with dotted-white lines indicating the plane used for 2D calculations. (**d**–**f**) Scale bars in are 100 nm and the colourbar represents the normalized electric field.

the effective refractive index of the gap plasmon mode (Supplementary Fig. 4). For each gap size $z$ we extract $Q_{LGP}$, the LGP resonance frequency $\omega_{LGP} = 2\pi c/\lambda_{LGP}$, where $c$ is the speed of light in vacuum (Fig. 2c), and the corresponding 3D fields (Fig. 2d,e). The computation reproduces experimentally observed $Q_{LGP} > 20$, remarkably, more than twice what has been attainable using dipolar antennas[21,22]. Separate two-dimensional eigenmode numerical analysis confirms that the LGP modes are standing-wave resonances formed by two counter-propagating fundamental gap plasmons (Fig. 2f)[23]. Thus, the LGP resonance condition is

$$\frac{2\pi}{\lambda_{LGP}(z)} n_{eff}(z) L_p + \varphi(z) = m\pi, \qquad (1)$$

representing a round-trip phase accumulation of $2\pi \cdot m$ along the prism length $L_p$, with a reflection phase $\varphi$[23,24], and mode order $m = 3$ in our case. The effective index for each $z$ is calculated using a corresponding input eigenfrequency of $\omega_{LGP}(z)$ determined at the same gap size from the 3D calculations. The LGP resonators are specifically designed to operate in the spectral region near 780 nm using this higher order mode, which has reduced coupling to radiation compared with the fundamental, and, therefore produces large quality factors. Given numerically calculated values for $n_{eff}(z)$ and $\lambda_{LGP}(z)$, the gap-dependent reflection phase $\varphi(z)$ is calculated using equation (1). It is important to note that the pNEMS can support hybrid dielectric-loaded surface plasmon travelling modes propagating parallel to the pad surface and through the effective waveguide formed by the SiN$_x$ beams. While small coupling to these modes must be contributing to the overall small radiation loss from our resonators, for the parameters used for our devices, such modes are not forming standing wave resonances strong enough to hybridize with the LGP and alter is spectral behaviour.

Using the explicit gap-dependence of $\lambda_{LGP}$, $n_{eff}$ and $\varphi$, we can derive a semi-analytical result for the optomechanical coupling strength $g_{OM} \equiv \partial \omega_{LGP}/\partial z \propto \partial \lambda_{LGP}/\partial z$ of the pNEMS. This parameter measures the strength of the interaction between the plasmonic and mechanical resonances in the pNEMS, and has a critical role in determining the ultimate transduction gain when using the pNEMS as a motion sensor. From equation (1) we find

$$g_{OM} = -\frac{c}{L_p n_{eff}} \left( \frac{m\pi - \varphi}{n_{eff}} \frac{\partial n_{eff}}{\partial z} + \frac{\partial \varphi}{\partial z} \right), \qquad (2)$$

whereby the optomechanical coupling arises from the gap-dependent effective index and reflection phase. Figure 3 shows the calculated $g_{OM}/2\pi$ as a function of gap size. The 3D values of $g_{OM}$ are extracted directly from the spectral shift in reflectivity with changing gap, whereas the semi-analytical curve is calculated using equation (2) and the values for $n_{eff}$ and $\varphi$ (Fig. 3 inset). The semi-analytical result allows separation of the two contributions, revealing that the rapid increase in $g_{OM}$ results from the increase in $n_{eff}$ with decreasing gap, being partially offset by the phase term (Supplementary Fig. 5). The reduced reflection phase with smaller gaps can be interpreted as the decrease in the effective resonator length due to the tighter longitudinal confinement of the LGP mode.

**Subdiffraction optical motion transduction.** Strong confinement and optomechanical coupling of LGP enables highly sensitive motion measurements of very small mass NEMS, such as our 600 fg cantilever. The LGP resonator resonantly enhances the phase and amplitude change of the far-field optical response while probing the motion from a nanoscale area, not limited by diffraction. Although a phase-sensitive measurement can be used, here we choose to detect the motion-induced LGP resonance shift

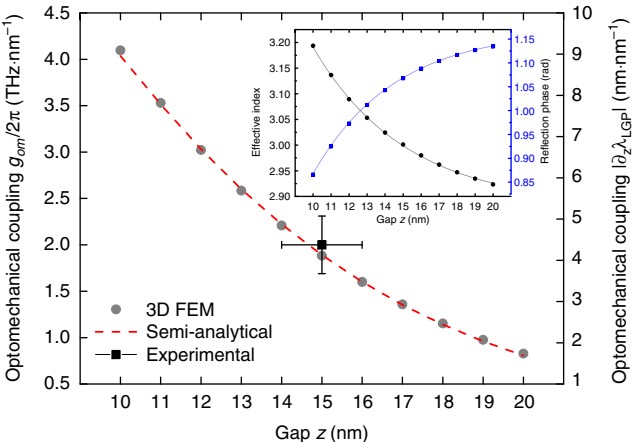

**Figure 3 | Calculated and measured optomechanical coupling.** Absolute values of the calculated optomechanical coupling constant $g_{OM}/2\pi$ (left axis) and $|\partial \lambda_{LGP}/\partial z|$ (right axis) from FEM modelled reflectivity data (grey circles) and from the semi-analytical model in equation 2 (dashed red line). The inset shows the two-dimensional calculated effective index (circles) and derived reflection phase (squares) as a function of gap; solid lines are fits to exponential functions. The black square shows the experimentally measured cantilever pNEMS optomechanical coupling value. Uncertainty indicates ± one standard deviation, corresponding to $(2 \pm 0.3)$ THz·nm$^{-1}$ for optomechanical coupling and $(15 \pm 1)$ nm for the full gap range consistent with AFM data for the gap. The error bar for the gap size is determined from the s.e. of the parabolic fit to AFM data (Supplementary Note 5), whereas the error bar for optomechanical coupling corresponds to the propagated error from all sources (Methods).

via an amplitude-modulated reflectivity signal. We measure the near-room-temperature thermally induced motion of the pNEMS, which are placed in a vacuum chamber to mitigate mechanical damping[6,25]. In a confocal system, a tunable laser is focused onto the sample through an optical window with a low-numerical aperture (NA = 0.3) objective (Methods, Supplementary Fig. 6). The probe wavelength ($\lambda_p$) is detuned from $\lambda_{LGP}$ to a fixed position near the maximum slope in the spectral reflectivity ($R$) to maximize the reflection amplitude modulation signal $\partial R/\partial z \propto Q_{LGP} g_{OM}$ (Supplementary Note 4).

Figure 4a,b show the thermal motion peaks in the spectra, well above the measurement background for both devices. The raw voltage signals, proportional to the vertical displacement of the prism, are calibrated by fitting the measured voltage power spectral density to a Lorentzian $S_{zz}(\omega)$ with an area given by the equipartition theorem (Methods, Supplementary Note 5, Supplementary Fig. 7) (ref. 25). From this procedure, we obtain the transduction gains $\alpha = (15.1 \pm 1.9)$ mV·nm$^{-1}$ and $(3.5 \pm 0.5)$ mV·nm$^{-1}$ and measurement noise floors $S_{zz,0}^{1/2} = (5.8 \pm 0.7)$ fm·Hz$^{-1/2}$ and $(23 \pm 2.5)$ fm·Hz$^{-1/2}$ for the cantilever and beam, respectively. For both devices, the same detector dark voltage noise corresponds to different values of the input-referred mechanical displacement noise reported in Fig. 4a,b. The different transduction gains determined for each device produce these differing values. The cantilever measurement noise floor is 1.5 orders of magnitude lower than comparable plasmonic systems[17], and approximately $4\times$ better than theoretically achievable using Doppler vibrometry on a large perfect reflector, shot noise limited at the same detected optical power ($P_0 = 45 \mu$W). However, pNEMS have the advantage that the high-precision motion transduction occurs from an area (the LGP mode footprint) $150\times$ smaller than the diffraction-limited, 3.2 μm diameter focal spot used to probe the system. The

measurement background is dominated, beyond detector dark noise, by photon shot noise.

The LGP motion transduction mechanism is verified using probe-wavelength-dependent measurements of motion on a third device: an 8 μm beam with LGP spectral reflectivity centered at 775 nm (Fig. 4c). Over the experimentally accessible wavelength detuning range $(\lambda_P - \lambda_{LGP})$, the strength of the signal, given by the dimensionless quantity $S_{zz}(\Omega_m)/S_{zz,0} - 1$, closely follows the $|\partial R/\partial\lambda|$ shape predicted from a Lorentzian fit to the LGP resonance of this device (Fig. 4d). Control experiments reveal both that the motion signal has the correct polarization dependence for the LGP mode and that the signal disappears when focusing the probe laser along the cantilever, away from the prism (Supplementary Note 6). These data further show that the LGP resonator is required to detect the motion of the devices.

The impressive performance of the pNEMS is attributed in part to the extremely large optomechanical coupling of the LGP modes, which we estimate in absolute value using the measured transduction gain $\alpha$ via

$$g_{OM} = \frac{2\pi c}{\lambda_P^2}\eta\left|\frac{1}{R_0}\frac{\partial R}{\partial\lambda}\right|^{-1}\frac{\alpha}{P_0 G_{DC}}, \tag{3}$$

where $G_{DC} = 25\,\text{mV}\cdot\mu\text{W}^{-1}$ is the measured photodiode gain, and $|\partial R/\partial\lambda|$ is the reflectivity slope evaluated at $\lambda_P$ from the spectral measurement in Fig. 2a; $R_0$ is the off-resonance reflectivity. The factor $\eta \approx 3.3$ accounts for a reduced $|\partial R/\partial\lambda|$ due to the

lower NA in the motion measurement system compared with the spectroscopic measurements (Methods, Supplementary Fig. 6)[26]. Using equation (3), we find optomechanical coupling constants $g_{OM}/2\pi = (2.0 \pm 0.3)\,\text{THz}\cdot\text{nm}^{-1}$ for the cantilever and $(0.6 \pm 0.1)\,\text{THz}\cdot\text{nm}^{-1}$ for the beam. The cantilever measured $g_{OM}$ is among the largest values reported to date in any optomechanical system, and agrees well with the $g_{OM}$ values expected for gaps ranging from 14 nm to 16 nm (Fig. 3). Using atomic force microscopy measurements of the cantilever shape (Supplementary Fig. 8), we estimate a 2 nm to 4 nm increase in the resonator gap size after release, which corresponds well to both the $g_{OM}$ values and a finite-element model of the device. Consistent with mechanical modelling results, the doubly clamped beam is expected to be deformed out of plane due to the residual stress, increasing the gap to greater than 20 nm and lowering $g_{OM}$.

## Discussion

Naturally, pursuing the largest value of $g_{OM}$ is of value for plasmomechanical systems, and we see that values in excess of $4\,\text{THz}\cdot\text{nm}^{-1}$ are in principle achievable by shrinking the gap to 10 nm or smaller. Beyond this point, classical electrodynamic theory breaks down in single-nanometre scale gaps[19], preventing further coupling and confinement improvements. Practically, due to the presence of attractive forces arising from Van der Waals interactions as well as possible residual embedded charges in the $SiN_x$, producing devices with as-deposited Cr layers thinner than

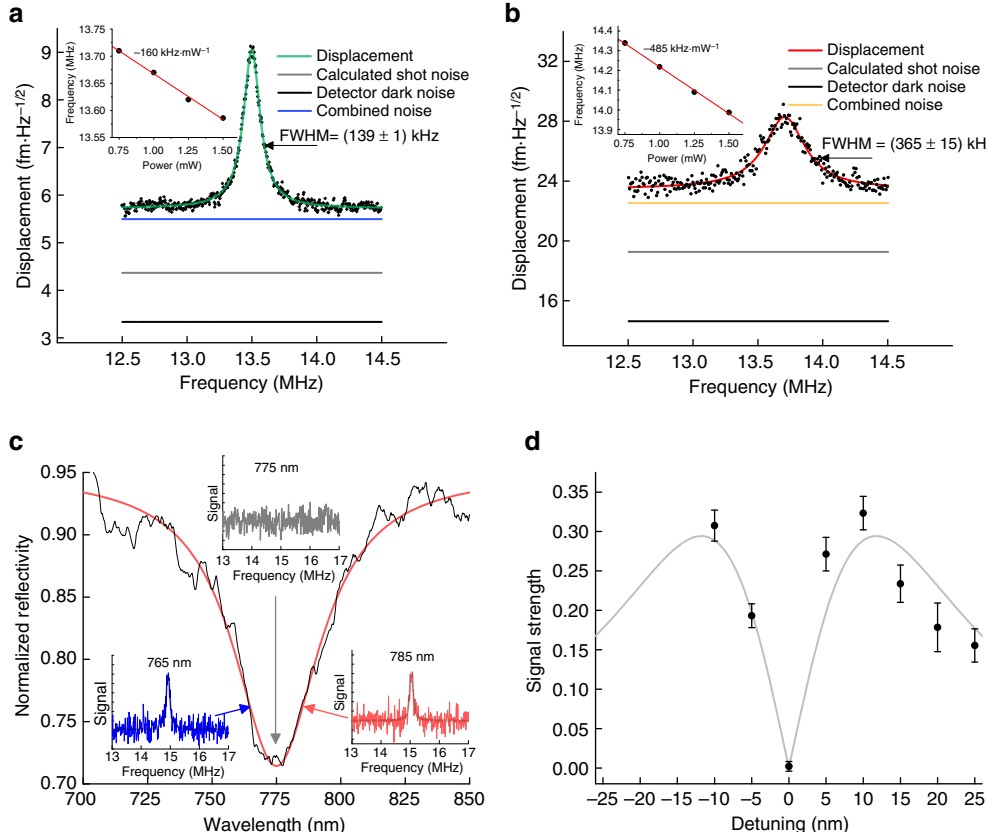

**Figure 4 | Motion transduction measurement and plasmonic origin verification.** Calibrated displacement spectrum for the (**a**) cantilever and (**b**) beam; insets show the measured mechanical frequency as a function of input power. (**c**) Spectral reflectivity for the third, beam device; insets show the motion power spectral density measured at each wavelength. (**d**) Measured dimensionless motion signal strength $S_{zz}(\Omega_m)/S_{zz,0} - 1$ as function of laser wavelength detuning. Signal-strength measurements are performed with an optical power of 1.25 mW delivered to the sample, which due to experimental limitations can only be achieved for detuning ranging from −10 nm to +25 nm. The grey line is the theoretical signal curve; error bars are the s.e. derived from Lorentzian fitting to the measured electrical power spectral density.

12 nm would require increased mechanical stiffness, which is undesirable for applications in sensing. Nevertheless, creating stiffer, faster and actuated pNEMS with sub-10 nm gaps is an attractive future avenue for producing electrically-tunable small plasmonic elements.

Figure 4 shows that varying the input power of the measurement laser causes a linear reduction in the mechanical frequency for both devices (insets, Fig. 4a,b). Similarly, we observe a change in the mechanical frequency of the third beam device as a function of detuning (Fig. 4c inset). In both cases, the behaviour likely does not result from the optical spring effect[8]. Instead, we suspect that bulk heating of the devices and a photothermal spring effect[27] contributes to the observed mechanical frequency changes. These intricate thermal effects will be the subject of a future study.

An advantage of the pNEMS architecture is that tailored thermal bimorph or electrostatic actuation can be included with additional patterning steps, before release of the devices, to achieve device tunability by electrical, as well as optical means. In contrast to photothermal effects, using optical forces for light-based control of pNEMS is somewhat more difficult to achieve. Intrinsic losses in plasmonic metals limit the plasmonic quality factor, yielding $g_{OM} \cdot Q_{LGP}$ products about two orders of magnitude lower than those in high-quality dielectric optomechanical systems with very large optical Q. Light-based control of plasmomechanical systems has only recently been demonstrated for extended structures[28], while observing such control in systems with single-element, highly localized interactions remains an open area of research.

We have introduced a new class of plasmonic resonators embedded into nanomechanical devices, and demonstrated them by measuring motion of sub-picogram cantilevers with an unprecedented combination of high sensitivity and small footprint. The low levels of input-referred mechanical motion noise in our motion measurements result from the unique features of our LGP resonators, namely the combination of extremely large optomechanical coupling strength, high plasmonic quality factor, and the large optical cross section of the LGP resonators, which increase the modulation of the reflected signal by small motion. The broadband nature of the LGP mode (full-width half maximum of 35 nm for both devices) can benefit chip-scale motion detection applications. Specifically, motion can be transduced from multiple devices, including device arrays, simultaneously using a single laser wavelength, despite small fabrication variations that produce device-to-device variation in $\lambda_{LGP}$. Moreover, inexpensive incoherent illumination sources, such as light-emitting diodes with spectral bandwidth filtered down to 10 nm or more, can potentially be used for these measurements in addition to lasers.

Beyond motion measurement, the dynamic LGP modes are attractive for future fundamental studies of nonlinear optomechanical coupling[28,29]. The pNEMS architecture admits arbitrary planar resonator and mechanical structure shapes (Supplementary Fig. 2), while precisely defining the large-aspect-ratio, nanoscale gap via a thin sacrificial layer. Therefore, the smallest lateral features (165 nm in this work) can be straightforwardly patterned using 193 nm photolithography for wafer-scale batch fabrication. This approach to creating plasmomechanical systems is backend-compatible with complementary metal oxide semiconductor (CMOS) processing. enabling future monolithic integration with high-speed multi-channel control circuits. Moreover, owing to large $Q_{LGP}$ and $g_{OM}$, only about 10 nm of electrostatic or thermal actuation is sufficient to shift the plasmonic resonance by one linewidth. Thus, pNEMS may enable not only large nanomechanical sensing assays, but also rapidly reconfigurable photonic devices, providing a path to realizing large-scale, randomly accessible photonic metamaterials.

## Methods

**Fabrication details.** A complete description of the fabrication process is given in Supplementary Note 1. Briefly, the process uses three steps of aligned electron beam lithography to first pattern the pads, then the prisms, both using a poly methyl methacrylate resist. The etch mask for creating the beams uses a commercial high-resolution resist. Electron-beam evaporation in combination with liftoff is used to deposit the metal layers. The structural device layer is formed using plasma-enhanced chemical vapour deposition of low-stress $SiN_x$ at 180 °C with a nominal 175 nm thickness. Reactive ion etching is used to pattern the mechanical devices. Wet-chemical etching of Cr is performed in a solution of ammonium ceric nitrate and the final release of the structures is performed using critical point drying in liquid $CO_2$.

**Spectroscopy measurements.** A supercontinuum laser spanning 500 nm to 850 nm is focused onto the sample with a 0.9 NA objective and reflected light is imaged onto a fiber-coupled spectrometer placed in a confocal arrangement. See Supplementary Note 2 for full details.

**Numerical modelling.** Numerical calculations are performed using a commercial finite element solver. Details are given in Supplementary Notes 3–5. 3D electromagnetic calculations are performed on a model comprising a 500 nm Si substrate, 45 nm Au pad, embedded $350 \times 165 \times 35$ nm$^3$ prism, a 175 nm thick, 1.25 μm wide nitride domain suspended above the Au by a variable air gap, and a top 250 nm air domain. A Gaussian beam with waist diameter 1.2 $\lambda$/NA with NA = 0.9 excites the domain, and reflectivity is calculated through the top entrance port. Plasmonic heating is calculated using the electromagnetic heat dissipation as a source for a thermal model of the same domain. The input field is a 0.3 NA focused beam with a power of 1.7 mW, representing experimental conditions during motion measurement. The two-dimensional eigenmode calculations take place on a plane with a normal aligned with the prism long-axis. Eigenfrequencies corresponding to $\omega_{LGP}(z)/2\pi$ are used to determine $n_{eff}$ and the gap plasmon wavevector. For the mechanical model, the cantilever or beam is attached to a 500 nm wide frame with the outside edges assumed to be rigidly clamped. The $SiN_x$ has elastic modulus $E_{SiNx} = 220$ GPa, Poisson ratio $v = 0.2$, and density $\rho = 2,200$ kg · m$^{-3}$, consistent with values reported in the literature for plasma-enhanced chemical vapour deposition grown films[30,31].

**Mechanical displacement calibration.** Considering a single (fundamental) pNEMS mechanical mode with generalized amplitude $q$, the motion power spectral density for $q$ is given by

$$S_{qq}(\omega) = \frac{4k_B T}{m_{eff,q}} \frac{\Gamma_m}{(\omega^2 - \Omega_m^2)^2 + (\omega \Gamma_m)^2}, \quad (4)$$

where $k_B$ is Boltzmann's constant, $\Gamma_m = \Omega_m/Q_m$, $\Omega_m$ is the mechanical angular frequency of the device fundamental vibration mode, $Q_m$ is the mechanical quality factor and $T$ is the temperature of the pNEMS device[26]. The effective mass $m_{eff,q} = \int \rho |\Phi|^2 \, dV/q^2$ is determined by the mode shape $\Phi$ normalized by $q$. For the generalized coordinate $q$, we use the vertical displacement $z$ at the prism location near the base, which is a factor $c_i = 0.1$ and 0.59 smaller than the maximum displacement at the cantilever tip and beam center, respectively (Supplementary Fig. 9). We compute $\Phi$ and $m_{eff,z}$ via an eigenfrequency finite-element calculation, and set $T$ to a slightly elevated value of 320 K, determined from modelling (Supplementary Fig. 10), to account for optical heating. We fit the measured voltage power spectral density $S_{VV}(\omega) = \alpha^2(S_{zz}(\omega) + S_{zz,0})$ with $Q_m$, $\Omega_m$, transduction gain $\alpha$, and the measurement noise floor $S_{zz,0}$ as adjustable parameters.

**Control experiments.** Details regarding control experiments can be found in Supplementary Note 6 and Supplementary Fig. 11.

**Uncertainty analysis.** All uncertainties reported are a single standard deviation unless noted otherwise. Uncertainty in devices sizes derives from the image calibration in the scanning electron microscope used for imaging, and is limited to less than 1% for device thickness based on pixel-to-metre calibration. The uncertainty in determination of the LGP wavelength and quality factor are determined from the Lorentzian fitting and are less than 1%. Uncertainty in the wavelength-dependent motion signal strength (Fig. 4d) is determined from Lorentzian fitting to the measured electrical power spectra. For both the experimentally determined optomechanical coupling and displacement noise floor values, the relative uncertainty is dominated by that of the calibration factor

$$\sigma_\alpha = \sqrt{(\partial_z \alpha)^2 \sigma_z^2 + (\partial_S \alpha)^2 \sigma_S^2 + (\partial_\Gamma \alpha)^2 \sigma_\Gamma^2}, \quad (5)$$

where $\partial_i \alpha$ and $\sigma_i$ are partial derivatives and uncertainties for $i = z_{rms}$, $S$, $\Gamma$ with $S = \sqrt{S_{VV,\Omega} - S_{VV,0}}$, and $\Gamma \equiv \Gamma_m$. Using the standard deviations derived from Lorentzian fits, we find that $\sigma_\Gamma$ and $\sigma_S$ are negligibly small. Thus, the uncertainty is dominated by contributions from thermal mechanical fluctuation amplitude $z_{rms} \equiv \langle z^2 \rangle \propto c_i \sqrt{T}/\sqrt{E_{SiNx}}$. Using $\sigma_E/E_{SiNx} \approx 8\%$ from nanoindentor measurements (one s.d. from six independent measurements), $\sigma_T/T \approx 7\%$

representing, conservatively, a maximum 20 K variability in the beam temperature above room 300 K, and $\sigma_c/c_i \approx 10\%$ from mechanical modelling, we find that $\sigma_\alpha/\alpha \approx 13\%$. The $g_{OM}$ values have an additional 10 % error resulting from the standard deviation for $G_{DC}$, determined from the linear fit.

**Data availability.** The authors declare that all data supporting this work are contained in graphics displayed in the main text or in the Supplementary Information. Data used to generate these graphics are available from the authors upon request.

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

## Acknowledgements

We thank D. Westly for assistance in developing the fabrication processes and A. Agrawal and J. Gorman for their technical suggestions and insightful comments on the manuscript.

## Author contributions

B.J.R developed the fabrication process, designed and performed the experiments, developed the computational models, analysed the data and wrote the manuscript. V.A.A developed the fabrication process, designed the experiments, developed the computational models and wrote the manuscript.

## Additional information

**Competing financial interests:** The authors declare no competing financial interests.

**Publisher's note**: 

