## [Peer Review File · Nature Communications]

Reviewers' comments:

Reviewer #1 (Remarks to the Author):

In this revised manuscript the authors have addressed all the issues i raised in my previous report. Therefore, i can now recommend this work for publication in Nature Communications.

Reviewer #2 (Remarks to the Author):

This is a revision submission of the manuscript titled "Nanomechanical motion transduction with a scalable localized gap plasmon architecture" by Brian Roxworthy and Vladimir Aksyuk. The authors report on the theoretical and experimental study of the coupling of localized gap plasmons and mechanical vibrational oscillations. In particular, the authors report on large optomechanical coupling strengths of around 2 THz/nm caused by the strong dependence of the effective index of the gap SPP on the insulating gap size. In addition, the authors report on interesting fabrication process to form nano-scale sized insulating gaps for localized gap plasmon nano-antennae.

The revised manuscript does not report on any extra experimental and theoretical data when compared to the initial submission. However, the authors do revise the manuscript following the referees' recommendations and comments. In particular, the explanatory paragraphs added to the manuscript make the paper clearer and more readable and gives somewhat better description of the novelty of the work.

The scientific and technical content of the manuscript is well explained and reasonable. The interpretations of the experimental results are supported by the theoretical/numerical data. The mythologies used in the work are clearly explained and valid. Essentially, the manuscript can in principle be interesting for the Nature Communications readership.

Some further recommendations can be found below:

- 1) Its not clear of how the strong optomechanical coupling can reduce the detection noise. The strong optomechanical coupling can increase the signal amplitude and consequently can improve the signal to noise ratio.
- 2) The authors emphasis the importance of the large optical bandwidth of the plasmonics. Please mentioned in the text of why the optical bandwidth is important for the targeted application areas.

Reviewer #4 (Remarks to the Author):

The authors demonstrate a platform for plasmo-mechanical devices that have high optomechanical coupling strength with sub-diffraction confinement. High sensitivity and low noise floors have been demonstrated in motion measurement. The manuscript is well written, and the authors have sufficiently answered the concerns from the previous reviewers. I suggest minor revision.

more comments.

1. In Fig2a, why the reflectivity of beam is smaller than cantilever? The inset of Fig2b shows larger gap leads to shorter λ_{LGP} and higher Q. Why the Q of beam is smaller than Q of Cantilever in Fig2a?

2. Was Fig2b calculated from simulation? or experimental results? It looks like simulation, but the authors say "The computation reproduces experimental $Q_{LGP} > 20$ (Fig2b, inset)"

3. In Fig3, at what wavelength the effective index and reflection phase were calculated? the measured optomechanical coupling value is $2\text{THz}\cdot\text{nm}^{-1} \pm 3.1\text{THz}\cdot\text{nm}^{-1}$. You sure the one sigma is $3.1\text{THz}\cdot\text{nm}^{-1}$? not 0.31?

4. Fig4a,b show the displacement spectra for cantilever and beam respectively. Can the authors explain why the noise in beam is much higher than that of cantilever? Shouldn't the detector dark noise be independent of the device under test?

5. Plotted in the inset of Fig4c, the mechanical resonance frequency is lower than 15MHz when probed with 765nm light, and it's higher than 15MHz when probed with 785nm light. Can the authors explain?

6. In the control experiment Sup Fig11a and Fig11b show different power and resonance frequency. Were the two measurements performed with different devices? Why? The author should be specific about the devices in control experiment.

Reviewer comments in bold; Author responses in plain text.

Reviewer #2

Its [sic] not clear of how the strong optomechanical coupling can reduce the detection noise. The strong optomechanical coupling can increase the signal amplitude and consequently can improve the signal to noise ratio.

We thank the reviewer for addressing this point of clarity and we agree with the statement. We tend to think about noise as input referred mechanical motion noise that corresponds to the actual experimental voltage noise. The strong g_{OM} (optomechanical coupling) increases the signal to noise ratio (SNR) of the motion detection, as indicated by Eq. S7 in the Supplementary Information. Larger SNR for voltage is due to the signal increase, as the reviewer points out, while in terms of motion, the thermal motion of the cantilever (the signal) stays constant, while the input-referred detection noise power density (in $\text{fm}/\text{Hz}^{-1/2}$) is decreased. Indeed, the large g_{OM} of the LGP modes is a key factor in obtaining the high motion sensitivity, corresponding to a minimum detectable (noise floor) value of $6 \text{ fm}\cdot\text{Hz}^{-1/2}$. We have modified the discussion on Page 6, Paragraph 2 from: “The low levels of noise in our motion measurements result from...the extremely large values of optomechanical coupling strength...” to the sentences given below, in order to more clearly explain this detail:

“The low levels of input-referred mechanical motion noise in our measurements result from...the extremely large values of optomechanical coupling strength...”

The authors emphasis [sic] the importance of the large optical bandwidth of the plasmonics. Please mentioned [sic] in the text of why the optical bandwidth is important for the targeted application areas.

A key benefit of the broadband nature of the plasmonic resonances is that it reduces problems associated with fabrication variations and can simplify instrumentation requirements for transducing motion of large arrays of devices. Both of these points are especially salient to building functional devices on the chip-to-wafer scale. In contrast, working with large- Q dielectric resonators requires sophisticated tunable and stabilized laser sources in order to produce and maintain a motion signal. Overall, the broadband response of the pNEMS factors into their overall effectiveness as a scalable motion sensing architecture.

We thank the reviewer for suggesting this clarification, and we have updated the discussion of the revised manuscript (Page 6, Paragraph 2) to more clearly highlight the benefit of the broadband LGP response:

“The broadband nature of the LGP mode (full-width half maximum of $\approx 35 \text{ nm}$ for both devices) can benefit chip-scale motion detection applications. Specifically, motion can be transduced from multiple devices, including device arrays, simultaneously using a single laser wavelength, despite small fabrication variations that produce device-to-device variation in λ_{LGP} . Moreover,

inexpensive incoherent illumination sources, such as light-emitting diodes with spectral bandwidth filtered down to ≈ 10 nm or more, can potentially be used for these measurements in place of lasers.”

Reviewer #4

In Fig2a, why the reflectivity of beam is smaller than cantilever? The inset of Fig2b shows larger gap leads to shorter λ_{LGP} and higher Q. Why the Q of beam is smaller than Q of Cantilever in Fig2a?

As discussed in the responses to Reviewer #2 above, the fabrication-induced device-to-device variation in the plasmonic resonances are not negligible. These manifest as variations in frequency and Q of the plasmonic resonances of nominally identical structures, and can be attributed to variations in as-deposited shape of the Au prism due to factors including Au grain formation, deposition inhomogeneities, and substrate or Au-pad thickness variations. Small process variations in the nitride film thickness, morphology, as well as height above the bottom pad may account for the observed variations in reflectivity off-resonance between devices. Optical beam alignment on the resonator and on the reflectivity reference device without a plasmonic resonator may also introduce some variation, which is equivalent to multiplication by a constant and does not change the visibility (modulation contrast) of the resonance.

The Q of the LGP resonances is determined from Lorentzian fits to the measured reflectivity data. We have determined that $Q_{LGP} = 23.0 \pm 0.6$ and 21.6 ± 1.7 for the cantilever and beam, respectively, and note that their difference is not very statistically significant. This indicates that both of the measured values of plasmonic Q are very close to the predicted values of from the electromagnetic model, given the uncertainties of the measurements.

We have updated the manuscript as follows to address this issue:

Page 3, Paragraph 2:

We have added the error in Q_{LGP} for both devices as determined from the standard error in the Lorentzian Fit.

Page 3, Paragraph 2:

We have added the following text:

“Differences in the measured linewidths and experimentally determined values of Q_{LGP} for the two devices, as well as their slight deviation from calculated values, are attributed to fabrication inhomogeneities such as variation in the prism shape.”

Was Fig2b calculated from simulation? or [sic] experimental results? It looks like simulation, but the authors say "The computation reproduces experimental $Q_{LGP} > 20$ (Fig2b, inset)"

Figure 2b shows the results of theoretical simulations of the gap-dependent LGP resonances, as stated in the main text (see Fig. 2b caption). We have modified the sentence as follows to more clearly state this point:

“The computation (Fig. 2b, inset) reproduces experimental $Q_{LGP} > 20\dots$ ”

In Fig3, at what wavelength the effective index and reflection phase were calculated? the measured optomechanical coupling value is $2\text{THz}\cdot\text{nm}^{-1} \pm 3.1\text{ THz}\cdot\text{nm}^{-1}$. You sure the one sigma is $3.1\text{THz}\cdot\text{nm}^{-1}$? not [sic] 0.31 ?

The values for the effective index and reflection phase in the Fig. 3 inset are calculated “on resonance”, i.e. at a unique wavelength/frequency for each gap size, corresponding to the λ_{LGP} (equivalently ω_{LGP}) determined from the 3D model. We have added the following sentence to Page 3, Paragraph 2, to more clearly highlight this detail:

“The effective index for each z is calculated using a corresponding input eigenfrequency of $\omega_{LGP}(z)$ determined from the 3D calculations at the same z .”

Indeed, the optomechanical coupling error should be $2\text{ THz}/\text{nm} \pm 0.31\text{ THz}/\text{nm}$. We thank the reviewer for pointing out this oversight and we have corrected it in the revised manuscript.

Fig4a,b show the displacement spectra for cantilever and beam respectively. Can the authors explain why the noise in beam is much higher than that of cantilever? Shouldn't the detector dark noise be independent of the device under test?

The noise figures presented for the different devices (as stated above, response to Reviewer #2, question 1) are the input-referred mechanical displacement noise values. In essence, the noise values are expressed in terms of a measured input quantity of interest, such as the calibrated mechanical displacement in this case. In terms of the raw voltage signal, as read from the electronic spectrum analyzer, the detector dark noise is the same for both devices: $\approx -92\text{ dBm}$ ($\approx 5.5 \times 10^{-8}\text{ V}\cdot\text{Hz}^{-1/2}$). The different values of the detector dark noise (in calibrated units of $\text{fm}\cdot\text{Hz}^{-1/2}$) result from the different values of the calibration factor/transduction gain present in the device. .

We have added the following sentence to Page 4, Paragraph 3 to address this subtlety:

“For both devices, the same detector dark voltage noise corresponds to the different values of the input-referred mechanical displacement noise reported in Figs. 4a,b. The different transduction gains determined for each device produce these differing values.”

Plotted in the inset of Fig4c, the mechanical resonance frequency is lower than 15MHz when probed with 765nm light, and it's higher than 15MHz when probed with 785nm light. Can the authors explain?

As stated in the text, the data for this figure is taken on a doubly clamped beam. For this geometry, the mechanical frequency change can be attributed to several contributing effects: (1) an optical spring effect, (2) constant bulk heating due to optical absorption and (3) a photothermal spring effect due to the dynamic coupling of optical absorption, thermal expansion and plasmonic resonance frequency shift. We have ruled out the optical spring effect, given that the relatively low plasmonic quality factor produces an average intracavity photon number of ≈ 5 – far too low to produce a significant enough optical force. We therefore attribute the observed behavior to thermal effects (2) and (3).

In the first case (2), optical absorption in the device increases temperature and adds compressive thermal stress in the doubly clamped geometry. This results in reduced out of plane stiffness and reduced resonance frequency as a function of increased absorbed optical power, regardless of the sign of the detuning of the laser from the plasmonic resonance. In the second case (3), the beam mechanical frequency has a dependence on the direction of the probe-laser detuning away from the plasmonic resonance through a well-known photothermal spring effect (see Ref. 26 in the revised manuscript). Briefly, a small downward motion of the beam (decreasing the gap) red-shifts the LGP resonance. For a red-detuned probe laser, the absorption and heating are increased. The resulting thermal expansion of the beam in our particular case is expected to cause an increase in the gap size, opposite to the original downward motion. Dynamically, this manifests as a positive-stiffness spring increasing the stiffness of the beam and increasing the frequency. Importantly, for a blue-detuned laser probe the effect has the opposite sign, decreasing the frequency.

The expected sign of the photothermal spring is therefore qualitatively consistent with the observed difference in the mechanical frequency between blue and red detuned cases. At present, however, we cannot draw definite conclusions from the observed behavior, as a more systematic study is required to distinguish and quantify the two described tuning effects. We are actively pursuing this phenomenon as the subject of a future study.

We have updated the manuscript on Page 6, Paragraph 2 to acknowledge the presence of this detuning-dependent mechanical shifts:

“Similarly, we observe a change in the mechanical frequency of the third beam device as a function of detuning (Fig. 4c inset). In both cases, the behavior likely does not result from the optical spring effect. [8] Instead, we suspect that bulk heating of the devices and a photothermal spring effect [26] contributed to the observed mechanical frequency changes. These intricate thermal effects will be the subject of a future study.”

In the control experiment Sup Fig11a and Fig11b show different power and resonance frequency. Were the two measurements performed with different devices? Why? The author should be specific about the devices in control experiment.

The data presented in Fig. S11 is collected from two cantilevers, selected at random, one for each control experiment. The purpose was to illustrate that all devices show the same behavior

indicative of the plasmonic motion transduction. The variation in the mechanical frequency between devices is consistent with the effects of variable clamping conditions, as described in Supplementary Section 5. In this case, the lower signal level for the “off-prism” data (red curve, Fig. S11a) is likely the result of scattering of the input beam from the edge of the cantilever, which can cause a drop in the confocally collected optical reflection signal from the device. In this particular case, it is likely that the beam was not translated perfectly parallel to the direction of the cantilever. For the polarization dependence measurement, the reduction in power between the two measurements is a result of an asymmetry between polarizations in the optical setup which causes more loss of the orthogonal polarization prior to introduction into the vacuum chamber. In this experiment, we did not correct for this slight power difference, resulting in the lower value of the red curve. However, we observe that for all powers tested up to the maximum of ≈ 3 mW introduced into the sample chamber, no motion signal was generated from any device in this orthogonal polarization.

Overall, the general behavior observed on all devices tested is consistent with what is shown in Fig. S11. Indeed, as part of the experimental testing, careful positioning and polarization of the probe laser is required to generate a visible motion signal as illustrated by Fig. S11.

To address these experimental subtleties, we have added the following text to the description of the control experiments in the Supplementary Information:

“All devices show similar behavior in these experiments, and we present data from two randomly selected cantilevers (one for each control experiment) from the same array as the device described in the main text. In the first experiment... thereby showing that the prism is required to transduce the pNEMS motion. The slight reduction in signal power for the off-prism data is caused by non-perfectly parallel translation of the probe laser with respect to the cantilever long axis, which reduced slightly the overall reflected optical power that is confocally collected.”

“In this case, the slightly lower signal power for the orthogonal polarization is caused by systematic reductions in the power delivered to the sample through our optical system. We nevertheless observe no motion transduction for this input polarization upon increasing the input power up to the instrumental limits of our system (≈ 3 mW). Notwithstanding these experimental subtleties, the observed behavior, i.e., elimination of the motion signal, is consistent with the data presented in Fig. S11 for all devices tested in this study.”

REVIEWERS' COMMENTS:

Reviewer #4 (Remarks to the Author):

The authors have addressed all my questions. I recommend this work for publication in Nature Communications.